# Effective Hydrogenation of 3-(2”-furyl)- and 3-(2”-thienyl)-1-(2’-hydroxyphenyl)-prop-2-en-1-one in Selected Yeast Cultures

**DOI:** 10.3390/molecules24173185

**Published:** 2019-09-02

**Authors:** Mateusz Łużny, Martyna Krzywda, Ewa Kozłowska, Edyta Kostrzewa-Susłow, Tomasz Janeczko

**Affiliations:** Department of Chemistry, Wrocław University of Environmental and Life Sciences, Norwida 25, 50-375 Wrocław, Poland

**Keywords:** biotransformation, chalcone, dihydrochalcone, yeast, sweeteners

## Abstract

Biotransformations were performed on eight selected yeast strains, all of which were able to selectively hydrogenate the chalcone derivatives 3-(2”-furyl)- (**1**) and 3-(2”-thienyl)-1-(2’-hydroxyphenyl)-prop-2-en-1-one (**3**) into 3-(2”-furyl)- (**2**) and 3-(2”-thienyl)-1-(2’-hydroxyphenyl)-propan-1-one (**4**) respectively. The highest efficiency of hydrogenation of the double bond in the substrate **1** was observed in the cultures of *Saccharomyces cerevisiae* KCh 464 and *Yarrowia lipolytica* KCh 71 strains. The substrate was converted into the product with > 99% conversion just in six hours after biotransformation started. The compound containing the sulfur atom in its structure was most effectively transformed by the *Yarrowia lipolytica* KCh 71 culture strain (conversion > 99%, obtained after three hours of substrate incubation). Also, we observed that, different strains of tested yeasts are able to carry out the bioreduction of the used substrate with different yields, depending on the presence of induced and constitutive ene reductases in their cells. The biggest advantage of this process is the efficient production of one product, practically without the formation of side products.

## 1. Introduction

In recent years, there has been growing interest from the food industry in sweeteners. Within this group of substances, dihydrochalcones are increasingly gaining attention. The increase in interest is due to the fact that dihydrochalcones are synthesized by plant cells, and they are a daily part of our diet [1,2]. They are present in citrus, apples, tomatoes, potatoes, bean sprouts, and other plants [3]. Dihydrochalcones have a wide spectrum of activity, such as antiviral (the ability to inhibit dengue virus proteases or herpes simplex virus), and anti-inflammatory and antioxidant activity [4,5,6,7]. They are also known for their activity against pathogenic microorganisms, including gram-positive and gram-negative bacteria, as well as fungi [8] and antimalarial or anti-tuberculosis activity [9]. Dihydrochalcone (phlorizin) is an active inhibitor of fungal tyrosinase [6,10]. Dihydrochalcones are also used in chemical synthesis to obtain biologically active compounds. 2′-Hydroxydihydrochalcon is used as a building block in the synthesis of propafenone—the active substance of anti-arrhythmic drugs [11,12,13].

Schallenberger’s hypothesis explains the sensation of sweet taste, according to which the flavor substance and the receptor create contact by hydrogen bonding [14]. This is due to the proton donor groups (AH) and an electron donor (B), which are also present in taste receptors and the structure of sweet substance. The sweetness of dihydrochalcones is related to their structure. The most important element of the structure of these compounds is the substitution of the B-ring in the meta or para position with at least one -OH group. The compounds that have three adjacent substituents on the B-ring do not have a sweet taste [15]. It is suspected that the structure of the B-ring also affects the perception of non-sweet aftertaste in dihydrochalcones, which do not always correspond to human preferences [16]. Commonly used in the food industry, neohesperidin dihydrochalcone is characterized by a licorice flavor and a cool feeling on the tongue [17]. There is also a group of dihydrochalcone analogs, that have a heteroatom in the B-ring and having non-sweet flavors [2]. This group also describes 3-(2”-furyl)-1-(2’-hydroxyphenyl)-propan-1-one (2), which, depending on the concentration, may have different taste sensations: A concentration of 0.01 ppm does not show any flavor properties; 1 ppm, soapy; umami, bitter; 10 ppm, licorice, soapy, slightly sweet, bitter, vegetable, terpenic; a particularly interesting impression is noted in higher concentrations – 100 ppm, bitter, licorice, light-sweet, celery, umami, and broth [2].

A method for obtaining 3-(2”-furyl)-1-(2’-hydroxyphenyl)-propan-1-one (2) with a conversion of 75%, as a result of the chemical synthesis of 2’-hydroxyacetophenone and furfuryl alcohol with NaOH, in the presence of an iridium catalyst, has been identified in [18]. The direct chemical hydrogenation of chalcones to dihydrochalcones requires the use of metal salts (complexes) of iridium, palladium, ruthenium, or nickel, and the entire reaction (due to its high flammability) must be conducted under strictly controlled conditions [19,20,21,22]. There are no reports in the literature regarding the use of biotechnological methods to obtain both, 3-(2”-furyl)- (2) as well 3-(2”-thienyl)-1-(2’-hydroxyphenyl)-propan-1-one (4).

The majority of living organisms are able to hydrogenate the double bond [23,24]. Yeasts possess their specific enzymes, capable of hydrogenating chalcones [9,25,26]. Such activity is also described in relation to other microorganisms, including bacteria, such as *Gordonia* sp. and *Rhodococcus* sp. [27], the entomopathogenic fungus, *Aspergillus flavus* [6], and the cyanobacterium, *Spirulina platensis* [28]. The reduction of activated alkenes by ente reductases [EC 1.3.1.31], which are flavoproteins from the old yellow enzyme (OYE) family, has been investigated in great detail [29,30,31]. The substrates in these reactions are predominantly hydroxy and methoxy derivatives of chalcones. In this study, we aimed to assess whether the heteroatom-containing substrate in the B-ring would also be accepted by the double-binding dehydrogenase, present in the tested biocatalysts.

The main aim of the study was to assess the capability of yeast strains for the biotransformation of a compound, containing a furan (**1**) and thiophene (**3**) substituent. Yeast strains, used in this experiment, are the subject of many years of research conducted in the Department of Chemistry of Wrocław University of Environmental and Life Sciences, regarding their catalytic capabilities [26,32,33], and have been selected as having a high ability in reduceing the double bond in the chalcone and its derivatives [26,27,28]. An additional goal was to optimize the biotechnological production of dihydrochalcone (**2**, **4**), which would enable the development of the method of obtaining the product on an increased scale.

## 2. Results and Discussion

Each of the eight tested microorganisms (*Rhodotorula rubra* KCh 4, *Yarrowia lipolytica* KCh 71, *Rhodotorula marina* KCh 77, *Rhodotorula rubra* KCh 82, *Candida viswanathii* KCh 120, *Rhodotorula glutinis* KCh 242, *Saccharomyces cerevisiae* KCh 464, *Candida parapsilosis* KCh 909) was able to transform both substrates into the expected products, but the efficiency of the described process differs significantly between the strains (Table 1 and Figure 1). High regioselectivity of the biocatalysts’ ability to reduce the double bond was observed, which was described before on the example of other compounds, e.g., chalcone containing no substituents and 2′-hydroxychalcone [32,34,35].

In the experiment, 3-(2”-furyl)- (**1**) and 3-(2”-thienyl)-1-(2’-hydroxyphenyl)-prop-2-en-1-one (**3**) were obtained by chemical synthesis, and then converted to 3-(2”-furyl)- (**2**) and 3-(2”-thienyl)-1-(2’-hydroxyphenyl)-propan-1-one (**4**) using biotransformation methods on a semi-preparative scale (Figure 2 and Figure 3). The obtained products **1**–**4** were purified (isolated yield > 70%) and their structures were determined on the basis of NMR analysis (^1^H-NMR, ^13^C-NMR and correlation spectra – HMBC; HMQC, COSY) as well as by gas chromatography (GC) and thin-layer chromatography (TLC) analysis (Appendix A).

The reduction of the double bond was the most efficient in the *Saccharomyces cerevisiae* KCh 464 strain, where over 99% of the product **2** was observed in the reaction medium after six-hour substrate incubation (Table 2). On the first day of the reaction, the four tested microorganisms exceeded the threshold of 90% of the substrate **1** conversion: *R*. *rubra* KCh 4, *R*. *rubra* KCh 82, *Y*. *lipolytica* KCh 71, *R*. *marina* KCh 77. In the culture of *C*. *viswanathii* KCh 120, the 90% limit was exceeded after three days of the biotransformation process. A surprisingly low conversion was observed for the strains *R*. *glutinis* KCh 242 and *C*. *parapsilosis* KCh 909 (Table 1). High hydrogenation of the double bond in the cultures of these strains was observed for chalcone and its methoxy derivatives [32,36].

Substrate **3** was most effectively transformed by the *Y*. *lipolytica* KCh 71 strain. The conversion above 99% was recorded after the third hour of biotransformation. A high efficiency of reduction of the double bond of chalcone derivatives was also confirmed for *Saccharomyces cerevisiae* KCh 464 strain (conversion > 98% after three hours of transformation). For most of the tested strains, a lower conversion of the substrate, containing thiophene (**3**) in its, compared to the furan-containing substrate (**1**), was observed. Interestingly, the strain *Rhodotorula rubra* KCh 4 after twelve hours showed only 10% of substrate **1** conversion, and after 24h it was already 91% (Figure 1), which may indicate that ene reductases are not present in the cell, but their production begins as a response to the stimulus (substrate induction) from the environment. An analogous induction of dehydrogenases (ene reductases) was observed for the strain *C*. *viswanathii* KCh 120. In the culture of this strain, substrate **3** was converted faster. However, conversion above 98% was achieved for both substrates at the same time in three days. Based on the obtained results, it can be concluded that the constitutive enzyme is present in *Y*. *lipolytica* KCh 71 and *S*. *cerevisiae* KCh 464 strains (observations for both substrates **1** and **3**). Biotransformations with these microorganisms show the fastest conversion progress, compared to other used microorganisms for both substrate **1** and **3** (Table 1, Table 2 and Table 3). Similar observations were obtained during the biotransformation of chalcone and described earlier [32]. 

The strains of the species *Saccharomyces cerevisiae* are widely described as biocatalysts, effectively reducing the double bond in various compounds – derivatives of chalcones [9], containing methyl, methoxy, hydroxy substituents [37], and even bromine or chlorine [33,38], in both A and B-rings. On the basis of the results obtained in this study, it can be concluded that a chalcone derivative, containing a heteroatom in the B-ring, is effectively transformed by this strain. In previous studies during biotransformation of chalcones, in addition to the hydrogenation product, a carbonyl reduction product was also observed [28,32,37,39]. In this study, additional products were not observed in any of the reactions carried out for the test compounds.

### Biotransformation

The obtained product 3-(2”-furyl)-1-(2’-hydroxyphenyl)-propan-1-one (**2**) was characterized by the following NMR spectrum: ^1^H-NMR (600 MHz) (CDCl_3_) δ (ppm): 3.07-3.12 (m, 2H, H-3), 3.34-3.41 (m, 2H, H-2), 6.06 (dq, 1H, *J* = 3.2, 0.8 Hz, H-3”), 6.29 (dd, 1H, *J* = 3.1, 1.9 Hz, H-4”), 6.90 (ddd, 1H, *J* = 8.2, 7.1, 1.2 Hz, H-5’), 7.02 (dd, 1H, *J* = 8.4, 0.9 Hz, H-3’), 7.32 (dd, 1H, *J* = 1.8, 0.8 Hz, H-5”), 7.49 (ddd, 1H, *J* = 8.5, 7.2, 1.6 Hz, H-4’), 7.92 (dd, 1H, *J* = 8.0, 1.6 Hz, H-6’), 12.23 (s, 1H, -O*H*). 

^13^C-NMR (151 MHz, CDCl_3_) δ = 22.50 (C-3), 36.73 (C-2), 105.69 (C-3”), 110.44 (C-4”), 118.72 (C-3’), 119.12 (C-5’), 119.40 (C-1’), 129.94 (C-6’), 136.56 (C-4’), 141.41 (C-5’’), 154.33 (C-2”), 162.56 (C-2’), 204.90 (C-1).

The obtained product 3-(2”-thienyl)-1-(2’-hydroxyphenyl)-propane-1-one (**4**) was characterized by the following NMR spectrum: ^1^H-NMR (600 MHz) (CDCl_3_) δ (ppm): 3.27-3.33 (m, 2H, H-3), 3.37-3.43 (m, 2H, H-2), 6.87 (dq, 1H, *J* = 3.4, 1.0 Hz, H-3”), 6.90 (ddd, 1H, *J* = 7.4, 7.1, 1.0 Hz, H-5’), 6.29 (dd, 1H, *J* = 5.1, 3.4 Hz, H-4”), 6.99 (ddd, 1H, *J* = 8.4, 1.1, 0.4 Hz, H-3’), 7.14 (dd, 1H, *J* = 5.1, 1.2 Hz, H-5”), 7.47 (ddd, 1H, *J* = 8.4, 7.3, 1.5 Hz, H-4’), 7.76 (dd, 1H, *J* = 8.1, 1.6 Hz, H-6’), 12.24 (s, 1H, -O*H*). 

^13^C-NMR (151 MHz, CDCl_3_) δ = 24.16 (C-3), 40.26 (C-2), 118.73 (C-3’), 119.13 (C-5’), 119.38 (C-1’), 123.72 (C-5”), 124.99 (C-3”), 127.06 (C-4”), 129.90 (C-6’), 136.60 (C-4’), 143.40 (C-2”), 162.56 (C-2’), 204.76 (C-1).

## 3. Materials and Methods 

### 3.1. Substrate

The substrates, used for biotransformation, were obtained by Claisen-Shmidt condensation reaction of 2-hydroxyacetophenone (**5**) with furfural (**6**) or aldehyde **7** [purchased from Sigma-Aldrich (St. Louis, MO, USA)] dissolved in methanol in an alkaline environment at high temperature (Figure 4 and Figure 5) according to the procedure described previously [32,40]. The resulting 3-(2”-furyl)-1-(2’-hydroxyphenyl)-prop-2-en-1-one (**1**) and 3-(2”-thienyl)-1-(2’-hydroxyphenyl)-prop-2-en-1-one (**3**) were used as a substrates for the biotransformation.

The resulting compound (**1**) was characterized by the following NMR spectral data: 

^1^H-NMR (600 MHz) (CDCl_3_) δ (ppm): 6.54 (dd, 1H, *J* = 3.4, 1.8 Hz, H-3”), 6.77 (d, 1H, *J* = 3.4 Hz, H-4”), 6.93 (ddd, 1H, *J* = 8.2, 7.2, 1.1 Hz, H-5’), 7.02 (dd, 1H, *J* = 8.4, 0.9 Hz, H-3’), 7.49 (ddd, 1H, *J* = 8.6, 7.2, 1.6 Hz, H-4’), 7.55 (d, 1H, *J* = 15.1 Hz, H-2), 7.56 (d, 1H, *J* = 1.4 Hz, H-5”), 7.68 (d, 1H, *J* = 15.2 Hz, H-3), 7.92 (dd, 1H, *J* = 8.1, 1.6 Hz, H-6’), 12.89 (s, 1H, -O*H*). 

^13^C-NMR (151 MHz, CDCl_3_) δ = 113.03 (C-4”), 117.29 (C-3”), 117.73 (C-2), 118.66 (C-3’), 118.98 (C-5’), 120.17 (C-1’), 129.77 (C-6’), 131.26 (C-3), 136.44 (C-4’), 145.55 (C-5”), 151.65 (C-2”), 163.66 (C-2’), 193.46 (C-1).

The resulting compound (**3**) was characterized by the following NMR spectral data:

^1^H-NMR (600 MHz) (CDCl_3_) δ (ppm): 6.95 (ddd, 1H, *J* = 8.1, 7.1, 1.1 Hz, H-5’), 7.02 (dd, 1H, *J* = 8.4, 1.2 Hz, H-3’), 7.12 (ddd, 1H, *J* = *J* = 5.0, 3.7, 0,3 Hz, H-4”), 7.41 (d, 1H, *J* = 3.6, Hz, H-3”), 7.44 (d, 1H, *J* = 15.2 Hz, H-2), 7.47 (d, 1H, *J* = 4.9 Hz, H-5”), 7.49 (ddd, 1H, *J* = 8.4, 7.2, 1.5 Hz, H-4’), 7.89 (dd, 1H, *J* = 8.1, 1.6 Hz, H-6’), 8.05 (dd, 1H, *J* = 15.2, 0,5 Hz, H-3), 12.85 (s, 1H, -O*H*).

^13^C-NMR (151 MHz, CDCl_3_) δ (ppm) = 118.74 (C-3’), 118.95 (C-2), 118.99 (C-5’), 120.06 (C-1’), 128.66 (C-4”), 129.65 (C-6’), 129.68 (C-5”), 132.89 (C-3”), 136.49 (C-4’), 138.00 (C-3), 140.30 (C-2”), 163.69 (C-2’), 193.27 (C-1).

### 3.2. Microorganisms

The studies were carried out on eight strains of yeasts of the species *Rhodotorula rubra* (KCh 4 and KCh 82), *Rhodotorula marina* (KCh 77), *Rhodotorula glutinis* (KCh 242), *Yarrowia lipolytica* (KCh 71), *Candida viswanathii* (KCh 120), *Saccharomyces cerevisiae* (KCh 464), and *Candida parapsilosis* (KCh 909), which have already been described [26,33], and come from the collection of the Department of Chemistry of Wrocław University of Environmental and Life Sciences, Poland. All the strains were cultivated on a Sabouraud agar consisting of aminobac (5 g), glucose (40 g) and agar (15 g) dissolved in 1 L of distilled water and pH 5.5 and stored in a fridge at 4 °C.

### 3.3. Analysis

Initial tests were carried out using TLC plates (SiO_2_, DC Alufolien Kieselgel 60 F_254_ (0.2 mm thick), Merck, Darmstadt, Germany). The product separation on a plate in cyclohexane is Ethyl acetate eluent (9:1 *v*/*v*). The product was observed (without additional visualization) under the UV lamp for the wavelength of 254 nm.

### 3.4. Gas Chromatography (GC)

GC analysis were performed using an Agilent 7890A gas chromatograph, equipped with a flame ionization detector (FID) (Agilent, Santa Clara, CA, USA). The capillary column DB-5HT (30 m × 0.25 mm × 0.10 µm) was used to determine the composition of the product mixtures. A temperature program was applied as follows: 80–300 °C, the temperature on the detector: 300 °C, injection 1 µl, flow 1 mL/min, flow H_2_: 35 mL/min, air flow; 300 mL/min, time of analysis; 18.67 min. The retention time of the substrate **1**—9,6 min, product **2** retention time—8,5 min. The retention times for compounds having the thiophene substituent **3** and **4** were recorded as 10,8 min, and 9,7 min, respectively (Appendix A).

### 3.5. NMR Analysis

NMR analysis was performed using a DRX 600 MHz Bruker spectrometer (Bruker, Billerica, MA, USA). The prepared samples were dissolved in deuterated chloroform CDCL_3_. The analyses performed, included ^1^H-NMR, ^13^C-NMR, HMBC (two-dimensional analysis) HMQC (heteronuclear correlation) and COSY (correlation spectroscopy) (Appendix A).

### 3.6. Screening

Erlenmeyer flasks with a capacity of 300 mL were used for biotransformation; each contained 100 ml of the culture medium – Sabouraud (3% glucose, 1% peptone). The transplanted microorganisms were incubated for three days at 24 °C on a rotary shaker (144 rpm). After this time, the substrates **1** and **3** were added in an amount of 10 mg, dissolved in 1 mL of DMSO (dimethyl sulfoxide). Samples were collected after 1, 3, 6, 12 hours and 1, 3, and 7 day of incubation. After this time, samples were extracted with ethyl acetate, dried with anhydrous magnesium sulfate (MgSO_4_), and analyzed by TLC and GC.

### 3.7. Semi-Preparative Scale 

Semi-preparative biotransformations were performed in 2 L Erlenmeyer flasks, each containing 500 mL of culture medium (3% glucose, 1% peptone). The transferred microorganisms were incubated for three days at 24 °C on a rotary shaker. After this time, the substrate was added in 100 mg, dissolved in 2 mL of DMSO. After ten days, the product in the mixture was isolated by triple extraction with ethyl acetate (3 extractions of 300 mL). After drying with anhydrous magnesium sulfate, and concentrating the sample by using a rotary evaporator, the obtained product was analyzed by GC and NMR.

The biotransformation product was separated on preparative TLC plates (1000 µm, silica gel plates (Anatech, Gehrden, Germany) with cyclohexane: Ethyl acetate eluent (9:1 *v*/*v*). After separation, the product was scraped from the plate, extracted twice from the gel with ethyl acetate, dried with magnesium sulfate and concentrated by a rotary vacuum evaporator. A one-day transformation of 3-(2”-furyl)-1-(2’-hydroxyphenyl)-prop-2-en-1-one (**1**) (100mg) in *Y*. *lipolytica* KCh 71 gave 72 mg (colorless oil) of 3-(2”-furyl)-1-(2’-hydroxyphenyl)-propan-1-one (**2**). One-day transformation of 3-(2”-furyl)-1-(2’-hydroxyphenyl)-prop-2-en-1-one (**3**) (100mg) in the same strain culture yielded 76 mg (pale yellow oil) of 3-(2”-thienyl)-1-(2’-hydroxyphenyl)-propan-1-one (**4**). The resulting products were then analyzed by NMR spectroscopy.

## 4. Conclusions

There is known catalytic activity of ene reductases present in many yeast species, consisting in the hydrogenation of chalcone to dihydrochalcone. In this study, 3-(2”-furyl)- (1) and 3-(2”-thienyl)-1-(2’-hydroxyphenyl)-prop-2-en-1-one (3) were biotransformed using eight yeast strains. The purpose was to test the microorganisms’ ability to selectively hydrogenate the double bond in chalcone containing heteroatom, in one of the rings, and select the most efficiently transforming strain. A significant factor in this type of reaction is not only its duration and the efficiency with which the substrate is transformed, but also the lack of by-products. Out of the available microorganisms, the fastest conversions of substrate 1 to the expected product were observed in *Saccharomyces cerevisiae* KCh 464 and *Y*. *lipolytica* KCh 71 strains. Product 2 was observed following the first hour of biotransformation (constitutive enzyme), and the conversion efficiency >99% was recorded as early as six hours after the start of the biotransformation. High conversion of substrate 1, higher than 90% after 24 hours of biotransformation, was also observed in the *R. rubra* KCh 4 culture. However, when analyzing the composition of the reaction mixture in twelve hours, the observed substrate conversion did not exceed 10%. This biotransformation pattern indicates that it is probably the result of substrate-induced ene reductases. Comparing these data to the conversion of 2-hydroxychalcone to dihydrochalcone, as previously described in [32], where the above-mentioned strains were also tested, we observed that the presence of the heteroatom in the B-ring practically does not affect the transformation efficiency—after 72 hours in the case of KCh 71 and KCh 464 strains; however, it is much more efficient in the case of the strain KCh 4. Substrate 3, which is composed of a sulfur atom in ring B, was analogously converted by the tested biocatalysts in the same way as substrate 1. Substrate 3 was most effectively transformed by *Y. lipolytica* KCh 71 strain culture. During this biotransformation, 3-(2”-thienyl)-1-(2’-hydroxyphenyl)-propan-1-one (4) was obtained in 99% yield after the third hour of transformation.

## Figures and Tables

**Figure 1 molecules-24-03185-f001:**
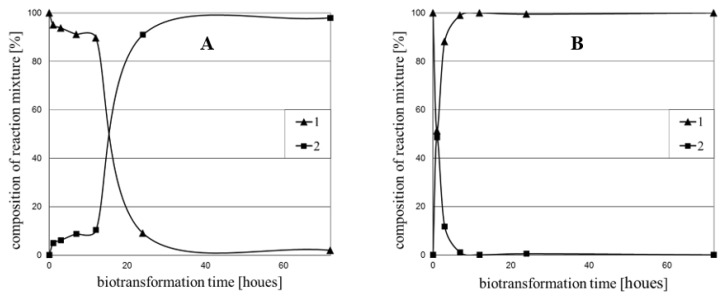
Time dependence of the transformation of chalcone (**1**) in the culture of: (**A**) *Rhodotorula rubra* KCh 4; (**B**) *Yarrowia lipolytica* KCh 71.

**Figure 2 molecules-24-03185-f002:**
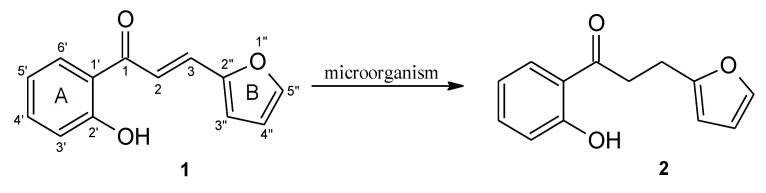
Biotransformation of 3-(2”-furyl)-1-(2’-hydroxyphenyl)-prop-2-en-1-one (**1**) by selected yeast cultures. The ring designations and the numbering of carbon atoms have been placed on compound **1**.

**Figure 3 molecules-24-03185-f003:**
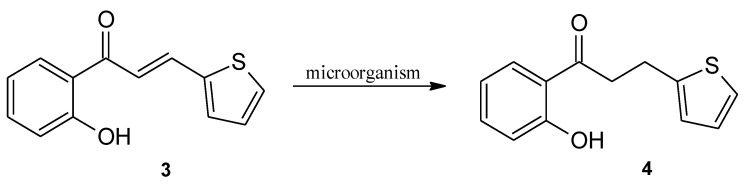
Biotransformation of 3-(2”-thienyl)-1-(2’-hydroxyphenyl)-prop-2-en-1-one (**3**) by selected yeast cultures.

**Figure 4 molecules-24-03185-f004:**
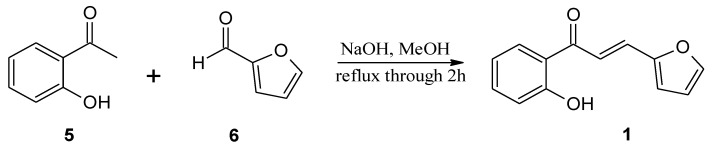
Synthesis of 3-(2”-furyl)-1-(2’-hydroxyphenyl)-prop-2-en-1-one (**1**).

**Figure 5 molecules-24-03185-f005:**
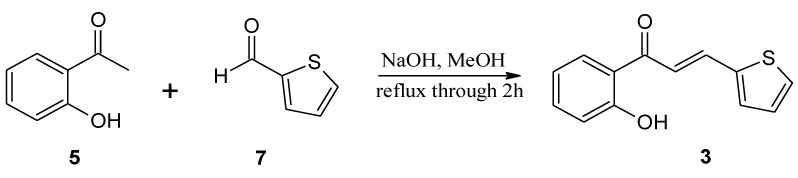
Synthesis of 3-(2”-thienyl)-1-(2’-hydroxyphenyl)-prop-2-en-1-one (**3**).

**Table 1 molecules-24-03185-t001:** Relationship between transformation of substrates **1** and **3** [%] and time in the cultures of tested strains.

Time [Days]	Strain Number																
	KCh 4		KCh 71		KCh 77		KCh 82		KCh 120		KCh 242		KCh 464		KCh 909	
	1	3	1	3	1	3	1	3	1	3	1	3	1	3	1	3
1	91	23	98	>99	92	38	98	59	49	84	0	52	98	98	2	9
3	99	33	>99	>99	>99	60	>99	84	>99	99	0	67	>99	99	14	12
7	>99	49	>99	>99	>99	63	>99	97	>99	99	17	87	>99	98	26	16

**Table 2 molecules-24-03185-t002:** Transformation of the substrate 1 by selected strains.

Time [h]	Strain Number			
	KCh4	KCh71	KCh120	KCh464
1	5 ± 1.1	52 ± 4.5	2 ± 0.5	40 ± 2.1
3	6 ± 0.1	88 ± 1.6	2 ± 0.4	84 ± 2.9
6	9 ± 0.5	>99 ±0.0	4 ± 0.9	98 ± 0.5
12	10 ± 0.9	>99 ± 0.0	9 ± 0.5	99 ± 0.6

**Table 3 molecules-24-03185-t003:** Transformation of the substrate 3 by selected strains.

Time [h]	Strain Number			
	KCh71	KCh120	KCh242	KCh464
1	67 ± 3.9	0 ± 0.0	1 ± 0.6	35 ± 3.6
3	>99 ± 0.0	10 ± 1.1	5 ± 0.5	64 ± 0.9
6	>99 ± 0.0	14 ± 1.2	9 ± 0.6	83 ± 0.6
12	>99 ± 0.0	25 ± 3.1	12 ± 0.9	98 ± 0.3

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
