# Peer review of "Effective Hydrogenation of 3-(2”-furyl)- and 3-(2”-thienyl)-1-(2’-hydroxyphenyl)-prop-2-en-1-one in Selected Yeast Cultures"

_molecules, 2019, doi:10.3390/molecules24173185_

Round 1
Reviewer 1 Report
This resubmitted manuscript has been improved a lot and includes more results on a new sulfur containing substrate as well as all the corrections/additions required by the reviewers.
There are a few more English language errors which need to be corrected (for example, line 126, quicklier must be replaced with faster).
Also, in the Materials and Methods, some editorial details must be corrected; for example a space is missing in most temperatures after the number (24 oC instead of 24oC).
I believe that in the present form this manuscript can be accepted for publication to Molecules after the above minor corrections.
Author Response
We would like to thank the Reviewer whose comments have enabled us to improve the paper from its original version. We hope that our explanations and corrections will be satisfying, and our paper will be finally accepted for publication.
Reviewer 1:
This resubmitted manuscript has been improved a lot and includes more results on a new sulfur containing substrate as well as all the corrections/additions required by the reviewers.
There are a few more English language errors which need to be corrected (for example, line 126, quicklier must be replaced with faster).
Response: The manuscript was once again carefully read and improved by a native speaker.
Also, in the Materials and Methods, some editorial details must be corrected; for example a space is missing in most temperatures after the number (24 oC instead of 24oC).
Response: Errors have been corrected as suggested.
I believe that in the present form this manuscript can be accepted for publication to Molecules after the above minor corrections.
Reviewer 2 Report
The resubmission of the manuscript “Effective hydrogenation of 3-(2”-furyl)- and 3-(2”-thienyl)-1-(2’-hydroxyphenyl)-prop-2-en-1-one in selected yeast cultures” describes now the biotransformation of two different chalcone derivatives by 8 different yeast strains.
The manuscript is improved with this resubmission, but there are still some important comment's:
The right use of the terms ene reductases, enoate reductases and old yellow enzyme are not always right. The big family of enzymes, that reduce double bounds, are the ene reductases this family has different subgroups. The most investigated subgroup is the old yellow enzymes group with FMN as cofactor. Another subgroup is the enoate reductases group with FAD and Fe-S Cluster. As long as you don’t know what enzyme catalyse the reaction you should write always ene reductase. For example line 70/71
Table 1/3 are this data from always one experiment or triplicates? Did you run controls? Improve the caption! Is this the found percentage of product, calculated by GC? And have have you calculated this with product standards? Because in your supplement Figure S27-S30 the product peak is 1h 16.03; 3 h 33; 6h 23.8 and 12h 132.145 and this corresponds not with your conversion data in Table 3 1 h 35 %; 3h 64 %; 6h 83% and 12h 98%. The product area goes down from 3 h to 6 h but your conversion is higher? How can be 23.8 peak area 83 % conversion and 132 peak area 98%? Or did you use internal standards for calculation?
The results are only a hint for a maybe substrate induced enzyme. You must weaken your sentence in line 286 with at least a maybe substrate induced ene reductase
And the results of the semi preparative scale experiments should be added with some sentence in the result section.
Author Response
We would like to thank the Reviewer whose comments have enabled us to improve the paper from its original version. We hope that our explanations and corrections will be satisfying, and our paper will be finally accepted for publication.
Reviewer 2:
The resubmission of the manuscript “Effective hydrogenation of 3-(2”-furyl)- and 3-(2”-thienyl)-1-(2’-hydroxyphenyl)-prop-2-en-1-one in selected yeast cultures” describes now the biotransformation of two different chalcone derivatives by 8 different yeast strains.
The manuscript is improved with this resubmission, but there are still some important comment's:
The right use of the terms ene reductases, enoate reductases and old yellow enzyme are not always right. The big family of enzymes, that reduce double bounds, are the ene reductases this family has different subgroups. The most investigated subgroup is the old yellow enzymes group with FMN as cofactor. Another subgroup is the enoate reductases group with FAD and Fe-S Cluster. As long as you don’t know what enzyme catalyse the reaction you should write always ene reductase. For example line 70/71
Response: In places where inaccuracies occurred, the enzyme descriptions were changed to "ene reductases" in lines: 25, 272, 286. However, the description enzymes performing the reduction of the double bond (line 70) is a description of the knowledge on the subject.
Table 1/3 are this data from always one experiment or triplicates? Did you run controls? Improve the caption! Is this the found percentage of product, calculated by GC? And have have you calculated this with product standards? Because in your supplement Figure S27-S30 the product peak is 1h 16.03; 3 h 33; 6h 23.8 and 12h 132.145 and this corresponds not with your conversion data in Table 3 1 h 35 %; 3h 64 %; 6h 83% and 12h 98%. The product area goes down from 3 h to 6 h but your conversion is higher? How can be 23.8 peak area 83 % conversion and 132 peak area 98%? Or did you use internal standards for calculation?
Response:
-Table 1 shows the results of the initial experiment checking the effectiveness of the biotransformation process of the tested strains. The main experiment was performed in one repetition. However, Tables 2 and 3 contain the average of three experiments and standard deviations made only for the strains carrying out the reduction most effectively.
-The control experiment was the incubation of test substrates in sterile medium without a biocatalyst. No substrate transformations was observed in this experiment.
-The caption have been improved
-The chromatograms in the annex are for demonstration purposes only. We chose these few chromatograms to show the rapid progress of the reaction and ideally selected conditions for the separation of substrates and products. We determined the conversion based on the percentage of substrate and product in the reaction mixture. The obtained results are a summary of three independent experiments. Different peak areas may be due to different amounts of solvents and injection sizes made in subsequent series (120 analyzes in one experiment). However, regardless of the sample size, the percentage of unreacted substrate and resulting product is identical.
The results are only a hint for a maybe substrate induced enzyme. You must weaken your sentence in line 286 with at least a maybe substrate induced ene reductase
Response: changed according to the suggestion
And the results of the semi preparative scale experiments should be added with some sentence in the result section.
Response: the results of the semi preparative scale experiments has been added to the discussion (lines 101-102)
This manuscript is a resubmission of an earlier submission. The following is a list of the peer review reports and author responses from that submission.
Round 1
Reviewer 1 Report
The manuscript titled: "Effective hydrogenation of 3-(furan-2”-yl)-1-(2’-hydroxyphenyl)-prop-2-en-1-onein selected yeast cultures" by Łużny et al. describes the biotransformation of a sole chalcone derivate by growing cells of 8 yeast strains.
The main result is an efficient and clean reduction of the double bond without byproducts, likely operated by enoate reductases. Moreover, the authors observed that the presence of oxygen as heteroatom in the B ring does not affect the transformation efficiency of the substrate compared to substrates tested in other studies previously reported.
The experimental structure is quite simple. The applied methods are mainly appropriate but not always exhaustively described. The results are not always clearly reported: in my opinion it is not clear if the described conversions are referred to the screening or to the semipreparative experiments. Conclusions are only partially supported by results, are in accordance with current knowledge, but novelty is limited
An additional check of the manuscript is required in order to clean many english errors.
I cannot suggest this paper for publication in the present form.
Major comments
If the purpose of the study was to assess the ability of the 8 selected strains to hydrogenate the double bond in heteroatom containing chalcones more than one substrate must be included in the study.
Were the strains selected (it is reported also in the title) or available? If they were selected, how? In section 3 citation n°32 does not help in obtaining this information and does not provide any description, remove and include the culture collection of origin.
For the authors, one of the most prominent results is the presence of constitutive/inducible enoate reductases in the 8 strains, but this observation was only deduced by the kinetics of the reduction, without no direct evidence (i.e. western blot data, specific oye activity). Moreover, no information about the growth kinetics of the different strains are reported and this parameter could affect also the rate of substrate reduction. The authors should confirm their hypothesis, for example, analyzing the enoate reductase activity in the crude cell extract of the strains grown in normal and in inducing condition comparing the specific activity.
Minor comments
Abstract…
20 … Also, when the scale was increased to semi-preparative, no side products were observed
This data was not described in result section.
1. Introduction
- Introduce the terms enoate reductase/ OYE
2. result and discussion
69-74 The aim of the study should be reported in the introduction
- Table 1 Improve the caption
- Figure 1 reports the same data in the table, it is redundant
- Where are the results of the semi-preparative scale reaction?
2.1 section is not necessary as separate section. Move figure 2 before in the text.
3. Material and methods
Substrate – The reported information does not allow to reproduce the synthesis.
Screening
Which are the solubilities of the substrate and the product? Are they completely dissolved in the culture media? Collected samples were representative of the whole reaction?
Semi-preparative scale
Which strains are tested? Include this information.
Reviewer 2 Report
This manuscript describes the screening of 8 selected yeast strains for their activity as ene-reductases using one specific substituted chalcone as substrate. The most productive and efficient strains in the reduction of the double bond have been determined.
Interestingly, no product derived from the carbonyl group reduction was detected. The conversion of biohydrogenation was excellent in many cases.
I have reviewed this manuscript and my comments are as follows:
1) In the Introduction as well as in the Discussion there is confusion about the B ring because it is not described clearly (see page 1 line 40, page 2 line 46, page 3 line 113).
2) Although the authors note clearly on page 1, lines 40-42 that the meta- or para- are the most important positions in the aromatic ring for -OH substitution, they have chosen to synthesize the OH-substituted chalcone in the ortho position. It would be necessary to synthesize also the meta and para isomers and screen them with the same strains.
3) Reference 19 is not correct and does not refer to the context on page 2.
4) In the semi-preparative scale the yield is missing.
5) In the Supplementary Materials session the GC analyses are missing
In conclusion, although this manuscript shows interesting results with one substrate, these results are limited and rather preliminary. As it stands right now, I don’t suggest publication to Molecules.
Reviewer 3 Report
The Article „Effective hydrogenation of 3-(furan-2“-yl)-1-(2’-hydroxyphenyl)-prop-2-en-1-onein selected yeast cultures” investigates eight different yeast strains and their potential for reducing the C=C double bond in the substrate.
First some general comments: The article has a lot of typing mistakes inside, for example mostly Brings instead of B-ring. Another general big disadvantage of this article is that often the citations didn’t fit to the text. Here only some examples:
Line 29 citation 7 is not about the impact of dihydrochalcones, it is only about the production of Dihydrochalcones.
Line 61 citation 22 has nothing to do with chalcones only with ketoisophorone.
The last really important thing the authors switch always the enzyme class, which should reduce the double bond. Sometimes they write that enoate reductase catalyse the reactions and sometimes Old yellow enzymes. But these are two completely different enzymes. Enoate reductases uses an iron-sulphur cluster and FAD/FMN and NADH for reduction, Old yellow enzymes are Tim barrel enzymes with FMN and NAD(P)H. Both enzyme groups belong to the big class of ene reductases. As long as not investigated which enzyme really catalyse the reaction please write always ene reductase to cover all.
Now to the results, the article contains only for all eight investigated yeasts one time course experiment for the conversion of the substrate. This experiment should be done in triplicates, to be sure that the results especially from strain KCh 4 are reproducible. Than the authors write in the material section that they have done preparative scale experiments, but this are not reported in the result and discussion section.